# Microneedle-Assisted Topical Delivery of Idebenone-Loaded Bioadhesive Nanoparticles Protect against UV-Induced Skin Damage

**DOI:** 10.3390/biomedicines11061649

**Published:** 2023-06-06

**Authors:** Yuan Xie, Jingping Ye, Yaqi Ouyang, Jianing Gong, Chujie Li, Yang Deng, Yang Mai, Yang Liu, Wenbin Deng

**Affiliations:** 1School of Pharmaceutical Sciences (Shenzhen), Sun Yat-sen University, Guangzhou 510275, Chinamaiy6@mail.sysu.edu.cn (Y.M.); 2School of Pharmaceutical Sciences (Shenzhen), Shenzhen Campus of Sun Yat-sen University, Shenzhen 518107, China

**Keywords:** UV-induced skin damage, idebenone, bioadhesive nanoparticles, microneedle

## Abstract

Ultraviolet (UV) radiation can penetrate the basal layer of the skin and induce profound alterations in the underlying dermal tissues, including skin pigmentation, oxidative stress, photoaging, glycation, and skin cancer. Idebenone (IDB), an effective antioxidant that suppresses melanin biosynthesis and glycation, can protect the skin from UV-induced damage, accounting for its use in commercial anti-aging formulations. Ideally, IDB formulations should retain IDB inside the skin for a sufficient period, despite disturbances such as sweating or swimming. Herein, we present an IDB topical formulation based on Tris (tris(hydroxymethyl)-aminomethane)-modified bioadhesive nanoparticles (Tris-BNPs) and microneedle-assisted delivery. We found that Tris-BNPs loaded with IDB (IDB/Tris-BNPs) effectively reached the basal layer of the skin and were retained for at least 4 days with a slow and continuous drug release profile, unlike non-bioadhesive nanoparticles (NNPs) and bioadhesive nanoparticles (BNPs) of similar sizes (ranging from 120–142 nm) and zeta-potentials (above −20 mV), which experienced a significant reduction in concentration within 24 h. Notably, IDB/Tris-BNPs showed superior performance against UV-induced damage relative to IDB/NNPs and IDB/BNPs. This effect was demonstrated by lower levels of reactive oxygen species and advanced glycation end-products in skin tissues, as well as suppressed melanogenesis. Therefore, the proposed IDB delivery strategy provided long-term protective effects against UV-induced skin damage.

## 1. Introduction

The sun is the primary source of natural ultraviolet (UV) radiation, which is categorized into three bands: UVA (315–400 nm), UVB (280–315 nm), and UVC (100–280 nm). UVA and UVB rays are well known to be damaging to the skin [1,2,3,4,5]. UVA radiation exposure causes the release of alpha-melanocyte-stimulating hormone (α-MSH) and the accumulation of melanin, leading to skin pigmentation and the synthesis of reactive oxygen species (ROS), resulting in oxidative stress and serious skin conditions such as wrinkles, inflammation, photoaging, and skin cancer [6,7,8,9]. Glycation has also been identified as a crucial factor in the accelerated aging of the skin following UV exposure [10]. Antioxidants have been reported to possess photoprotective characteristics, capable of mitigating skin pigmentation, glycation, oxidative stress, and photoaging. Consequently, there is widespread interest in the development of formulations containing antioxidants for daily protection against UV-induced damage [7,11].

Idebenone (IDB), a synthetic analog of coenzyme Q10 (CoQ10), has been demonstrated to possess free radical-scavenging activity. IDB is a more effective antioxidant relative to several others that are commonly used in skincare products, including vitamin C, vitamin E, and coenzyme Q10 [12,13]. However, the use of IDB as an antioxidant is limited by its stability [14,15]. Systemic administration of IDB may result in various side effects that are potentially severe and far-reaching [16]. To mitigate the long-term toxicity associated with IDB, it is generally recommended to adjust dosages and treatment plans or use targeted delivery methods. Efficient skin permeation and sustained skin retention of the active ingredient are crucial for achieving the intended therapeutic effect.

In a previous study, we synthesized bioadhesive nanoparticles (BNPs) from non-bioadhesive nanoparticles (NNPs) composed of polylactic acid-hyperbranched polyglycerol (PLA-HPG) polymer [17]. The benefits of BNPs have been demonstrated in various applications, such as sunscreen formulations, where they enhance skin adhesion, reduce systemic toxicity, and improve the retention of chemotherapeutics after intraperitoneal and intratumoral administration. In addition, BNPs have been demonstrated to retain antivirals when administered intravaginally [17,18,19,20]. The benefits of BNPs are attributed to the abundance of aldehydes on their surface. These aldehydes covalently bind to amine groups in biomolecules found in the stratum corneum (SC), on cell surfaces, within the protein-rich matrix of tumors, and in extracellular spaces [21]. However, upon topical administration, BNPs are predominantly localized on the surface of the SC, preventing them from penetrating deeper layers. In a prior study, Tris (tris(hydroxymethyl)-aminomethane)-modified BNPs (Tris-BNPs) were obtained, using Tris to modify the BNPs so that the bioadhesivity of Tris-BNPs was temporarily masked, allowing them to permeate the SC of psoriatic skin lesions [22]. Nonetheless, permeability of Tris-BNPs was not observed on healthy skin. Given that melanocytes in the epidermis are typically located in the basal layer above the basement membrane, we hoped that Tris-BNPs could also be applied to normal skin and successfully penetrate the basal layer of skin lacking the high permeability characteristics observed in psoriasis [23]. With the aid of microneedles, we tried to overcome the limitations of the SC of healthy skin. The advantages of the widely used microneedle-assisted transdermal drug delivery strategy include the avoidance of hepatic and gastrointestinal metabolism [24]. A recent study has shown that ex vivo transdermal delivery of nicotinamide mononucleotide (NMN) using polyvinyl alcohol (PVA) microneedles can improve the bioavailability of NMN [25].

To ensure long-term protection of skin tissues from UV-induced damage, we aimed to retain IDB inside the skin for a particular period, despite disturbances such as sweating or swimming. In this study, Tris-BNPs loaded with IDB were administered using microneedles into the skin of healthy mice to assess their protective effect against UV-induced skin damage (Figure 1). Tris-BNPs loaded with IDB (IDB/Tris-BNPs) were obtained using Tris to modify BNPs loaded with IDB (IDB/BNPs). The IDB/BNPs were obtained by oxidizing IDB-encapsulated NNPs (IDB/NNPs) using NaIO_4_ (Figure 1A). Instead of instant adhesion, with a novel design, IDB/Tris-BNPs were diffused from the skin surface to the basal layer similarly to IDB/NNPs via the channels produced by the microneedles. Furthermore, the Tris coating gradually diffused during the penetration process, resulting in the exposure of aldehyde groups in BNPs that could react with amine groups in the basal layer cell membrane protein. The reaction ability between aldehyde groups in BNPs and amine groups has been proven previously [26]. Thus, the Tris-BNPs exhibited excellent bioadhesivity, as for BNPs, and firmly adhered to the skin, where they were retained longer (Figure 1B,C). The microneedle-assisted topical delivery of Tris-BNPs provides a promising platform for skin protection and disease treatment.

## 2. Material and Methods

### 2.1. Materials

Dimethyl sulfoxide (DMSO), ethyl acetate, diethyl ether, and Na_2_SO_3_ were purchased from Guangzhou Chemical Reagent Factory. NaIO_4_ and polylactic acid (PLA, COOH terminated, MW = 15.0 kDa) were ordered from Aladdin and Shunna Biotechnology, respectively. IDB was supplied by Maya Co., Ltd. (Maya Co., Ltd., Coventry, UK) Tris-HCl was from Biosharp. Cy5-NH_2_ was obtained from Lumiprobe Corporation (Lumiprobe Corporation, Cockeysville, MD, USA). Dialysis tubes (MWCO (molecule weight cut-off) 0.5–1 kDa; Float-A-Lyzer G2) were purchased from Spectrum Labs (Spectrum Labs, San Francisco, CA, USA).

### 2.2. Manufacture of Hyperbranched Polyglycerol (HPG)

Anionic polymerization was used to synthesize HPG. In brief, 4.67 mmol of 1,1,1-trihydroxy-methylpropane (THP) and 1.4 mmol of KOCH_3_ were added to a flask prefilled with argon heated to 95 °C in an oil bath. The system was placed under vacuum for 10 min after being connected to a vacuum pump. The argon was regularly replenished in the system.

### 2.3. Synthesis of PLA-HPG and PLA-Cy5

#### 2.3.1. PLA-HPG

To 5 g of PLA pre-dissolved in dichloromethane (DCM), 2.3 g of HPG dissolved in dimethylformamide (DMF) was added. The resulting mixture was dried using a molecular sieve. Next, 13.5 mg of 4-(N, N-dimethylamino) pyridine (DMAP) and 0.08 mL of diisopropyl carbodiimide (DIC) were added to the reaction mixture, which was continuously stirred for 5 days at room temperature. The final product was obtained as precipitates by pouring the reaction mixture into cold ether, which was later collected by centrifugation. The precipitates were redissolved in DCM, and the product was precipitated with the use of cold ether. Finally, the polymer underwent centrifugation and was subjected to vacuum drying for 48 h.

#### 2.3.2. PLA-Cy5

PLA-Cy5 was synthesized via a condensation reaction involving the amine group of Cy5-NH_2_ and the carboxylic acid group of PLA. Briefly, 15 mg of Cy5-NH_2_ and 0.02 mL of DIC were added to a solution containing 1.95 g of PLA dissolved in DCM. The mixture was stirred for 24 h at room temperature. The final product was precipitated by adding cold diethyl ether, followed by centrifugation at 4500 rpm for 15 min. The final product was then vacuum dried for 2 days.

### 2.4. Preparation of NPs

An amount of 22.5 mg of PLA-HPG copolymer was dissolved in 0.85 mL of ethyl acetate:DMSO (*v*:*v* = 10:7) solution. Next, 2.5 mg of PLA-Cy5 or IDB was added to the PLA-HPG solution to prepare dye-loaded or drug-loaded NPs. The resultant mixture was added to 2 mL of deionized (DI) water while vortexing. The mixture was sonicated thrice (10 s/sonication) using a probe at 162.5 W. The emulsion was diluted with 10 mL DI water while stirring. The ethyl acetate was evaporated using a rotary evaporator. Using Amicon filters with 100 kDa MWCO, the NP solution was washed twice, before being redissolved in DI water. Before usage, the NPs were stored at −20 °C.

NNPs were oxidized to BPNs using NaIO_4_ solution. NNPs (25 mg/mL) were incubated with three volumes of 0.1 M NaIO_4_ for 5 min, and the reaction was then quenched with three volumes of 0.2 M Na_2_SO_3_. The resultant BNPs, with or without PLA-Cy5 or IDB, were washed twice with DI water using Amicon filters, before being redissolved in DI water to remove any residue organic solvent or free drug. The final step was to mix the BNPs with Tris solution to generate Tris-BNPs.

### 2.5. Characterization of Various NPs

The hydrodynamic size and zeta potential of the NPs were measured using a Zetasizer Nano ZS (Malvern Instruments, Malvern, UK). The morphology of NPs was visualized by TEM (FEI, Tecnai G2 F30). On the carbon-supported copper grid, a drop of the NPs suspension was applied, followed by a drop of phosphotungstic. A filter paper was then used to remove most of the drops. A plate reader that emitted fluorescence at the excitation/emission wavelengths of 650/680 nm was employed to determine the concentration of Cy5 within the NPs.

### 2.6. In Vitro IDB Release Profile

The release of IDB from Tris-BNPs, NNPs, and BNPs was analyzed by incubating 0.5 mg/mL of IDB/Tris-BNPs, IDB/NNPs, or IDB/BNPs with 0.1% sodium dodecyl sulfate (SDS) in phosphate-buffered saline (PBS) at 37 °C in a shaker (Bluepard, Shanghai, China). The samples were centrifuged using Amicon Ultra-0.5 mL filters (100 kDa MWCO) at 0, 6, 24, 48, 72, and 96 h, and the filtrate was collected for high-performance liquid chromatography (HPLC) analysis on a C18 column (4.6 × 250 mm, particle size: 5 μm). The samples were examined at a wavelength of 278 nm. The mobile phase was composed of methanol:ddH_2_O (72:28). The injection volume and flow rate were 20 μL and 1.0 mL/min, respectively.

### 2.7. In Vitro Cytotoxicity Evaluation of NPs

The cytotoxicity of NNPs, BNPs, Tris-BNPs, IDB-loaded NPs (IDB/NNPs, IDB/BNPs, and IDB/Tris-BNPs), and free IDB was evaluated in mouse embryonic fibroblast cells (NIH/3T3 cells), B16 melanoma F10 cells (B16F10 cells), and human keratinocyte epithelial cells (HaCaT cells) (all from Procell Life Science & Technology, Wuhan, China). HaCaT (ATCC FS-0241) and NIH/3T3 cells were cultured in Dulbecco’s modified Eagle medium (DMEM) (Gibco, Grand Island, NY, USA) that contained 10% fetal bovine serum (FBS, Gibco) and 1% penicillin/streptomycin. B16F10 cells were cultured in Roswell Park Memorial Institute (RPIM) 1640 medium (Solarbio, Beijing, China) supplemented with 10% FBS and 1% penicillin/streptomycin.

NIH/3T3 and HaCaT cells were cultured in a 96-well plate at a density of 10,000 cells/well, with each well containing 100 μL of culture medium. B16F10 cells were plated at a density of 5000 cells/well. The cells were then incubated overnight at 37 °C. The next day, the medium was removed. A volume of 100 μL of DMEM, NNPs, BNPs, Tris-BNPs, IDB/NNPs, IDB/BNPs, IDB/Tris-BNPs, or free IDB in DMEM solution was added to the cells. The cells were incubated at 37 °C for 24 h. A CCK-8 assay kit (Dojindo, Kumamoto, Japan) was used to assess cell viability.

### 2.8. Melanogenesis Assessment of NPs

The melanin content of cells was evaluated following previously described methods [27]. Briefly, B16F10 cells were seeded in a 6-well plate at a density of 12 × 10^4^ cells, and 20 μL of α-MSH (100 mM) was added to each well. This was followed by overnight incubation of the cells at 37 °C under 5% CO_2_. The medium was subsequently removed. After washing with 1 mL 1× PBS, 2 mL of DMEM and 10 μL α-MSH (100 mM) were added to the cells, which were subsequently incubated at 37 °C under 5% CO_2_ for 48 h. The cells were digested using trypsin, terminated by the addition of DMEM, and centrifuged at 1000 rpm for 5 min. The sediment was resuspended in 1× PBS and centrifuged at 1000 rpm for 5 min. After adding 300 μL of 1 M NaOH solution containing 10% DMSO to the sediment, the mixture was heated in a water bath maintained at 80 °C for 1 h. Melanin standard solutions with concentrations of 2, 1, 0.5, 0.25, 0.125, 0.0625, 0.03125, and 0.015625 mg/mL were prepared. The absorbance values of the melanin standard solutions were measured at 405 nm using a microplate spectrophotometer. A standard curve was plotted to determine the melanin content in the epidermis of each mouse.

### 2.9. Retention and Penetration Analysis of Various NPs

All animal procedures were conducted according to protocols that were approved by Shenzhen Bay Laboratory Biotechnology, Shenzhen, China (Ethic Committee Name: Shenzhen Bay Laboratory Animal Ethics Committee and Approval Code: AELIUY202201). The animals were housed at the Shenzhen Bay Laboratory, Shenzhen, China. Various NPs loaded with 1.0 mg/mL of PLA-Cy5 were topically administered to the dorsal skin of C57BL/6 mice to assess the retention of the various NPs on the skin, when administered with or without microneedles (Chenye, Ganzhou, China, medical device certification No. 20210014). Photographs of NPs retained on the skin at different time points were taken using an in vivo imaging system (IVIS) Lumina XR (Perkin Elmer, Beaconsfield, UK) based on excitation and emission wavelengths of 650 nm and 680 nm. The mice were euthanized after the retention test. The dorsal skins of the mice were harvested and frozen in optimal cutting temperature compound (OCT) to assess the degree of penetration. Next, 10 μm sections of the frozen tissues were obtained, mounted on glass slides, and observed under a fluorescence microscope (Eclipse Ti2-U).

### 2.10. Establishment of an Artificial Tanning Model

A solar simulator with a xenon arc lamp and filters (HPA400/30S, Philips, Hamburg, Germany) was used as the UV radiation source. The dorsal skins of C57BL/6 mice were shaved, and the mice were housed in cages (containing a maximum of five mice per cage) maintained at 25 °C. In addition, a 12 h light–dark cycle was observed. On day 1, the solar simulator was used for 1 min to deliver UV radiations between 290 nm and 400 nm to all mice, except those in the no UV control group. Then, 5 mg/mL of NNPs, BNPs, IDB/Tris-BNPs, blank Tris-BNPs, or commercial products (equivalent IDB dose) were topically applied to the dorsal skin with microneedles scrolling back and forth 10 times. All mice were euthanized on day 4. The dorsal skin was isolated on ice for subsequent tests. The chroma values representing skin color were measured using IMATE M6601 each day. Three measurements were taken for each parameter, and the average value of each measurement was used for the analysis.

### 2.11. Skin-Protecting Activity of NPs

The melanin content in the epidermis of the mice in each group was determined using a previously described method [28]. An amount of 200 mg of the epidermis sample and 1 mL of tissue lysate (10× PBS solution containing 1% Triton X-100 and 1 mM phenylmethylsulfonyl fluoride [PMSF]) were mixed. The resulting mixture was homogenized for 15 min at 4 °C and then centrifuged (at 16,000× *g* for 5 min at 4 °C). After removing the supernatant, the sediment was vortexed while adding 200 μL of the washing solution (ethanol:ethyl ether, *v*:*v* = 1:1) and centrifuged at 16,000× *g* for 5 min at 4 °C. After removing the supernatant, 200 μL of 1 M NaOH solution containing 10% DMSO was added and heated in a water bath at 80 °C for 1 h. Melanin standard solutions with concentrations of 2, 1, 0.5, 0.25, 0.125, 0.0625, 0.03125, and 0.015625 mg/mL were prepared. The absorbance values of the melanin standard solutions were measured at 405 nm using a microplate spectrophotometer. A standard curve was plotted to determine the melanin content in the epidermis of each mouse.

An amount of 1 mL of 1× PBS was used to fully homogenize a 100 mg sample of the epidermis. The homogenized mixture was centrifugated at 10,000× *g* for 5 min at 4 °C. After that, 190 µL of the supernatant and 10 µL of dichloro-dihydro-fluorescein diacetate (DCFH-DA, Beyotime, Nantong, China) were mixed and placed in a 96-well plate. The plate was then incubated in the dark for 30 min at 37 °C. An amount of 50 μL of the supernatant was diluted 90 times with 1× PBS. Next, 100 μL of the resulting mixture was tested with a bicinchoninic acid (BCA) assay kit (Beyotime) to quantify the protein levels. A microplate reader (Tecan, Infinite E plex) was used to detect the fluorescence signals. The excitation and emission wavelengths were set at 488 nm and 525 nm, respectively).

AGEs content was measured using a Mouse AGEs ELISA kit (Mlbio, Charlotte, NC, USA, ml002154). An amount of 200 mg of epidermal sample and 1 mL of tissue lysate (10× PBS solution containing 1% Triton X-100 and 1 mM PMSF) were added, homogenized at 4 °C for 15 min, and centrifuged. Next, 50 μL of the supernatant and 100 μL of horseradish peroxidase (HRP)-labeled detection antibody were added to a 96-well plate and incubated for 60 min at 37 °C. The 96-well plate was then dried. Then, 350 μL of washing solution was added to each well. The plate was left to stand for 1 min before the washing solution was discarded; this process was repeated five times. Substrates A and B (50 μL) were added to each well and incubated at 37 °C in the dark for 15 min. To quench the reaction, 50 μL of a termination solution was added. Within 15 min of quenching the reaction, a spectrophotometer (Bio Tek, Ahmedabad City, India) was used to measure the optical density (OD) at 450 nm.

### 2.12. Histopathological Study

IDB-loaded NPs or commercial products were applied to the dorsal skin of mice. After euthanasia on day 4, the major organs (heart, liver, spleen, lung, kidney, and brain) were collected and fixed in 4% paraformaldehyde (PFA) overnight. The tissues were then decalcified, embedded in paraffin, sectioned, and stained with H&E (Servicebio Co. Ltd., Shenzhen, China). All stained paraffin sections were visualized under an optical microscope.

### 2.13. Statistical Analysis

All experiments were performed at least thrice, and the results were shown as mean ± standard deviation (s.d.). To analyze the differences among the different groups, a one-way analysis of variance (ANOVA) was performed, followed by Tukey’s post hoc analysis using GraphPad Prism 9.3.1(350) software. The significance level was depicted on plots as follows: ns (not significant), *p* > 0.05; * *p* < 0.05; ** *p* < 0.01; and *** *p* < 0.001.

## 3. Results and Discussion

### 3.1. Characterization of IDB/Tris-BNPs

NNPs composed of PLA-HPG polymer were oxidized using NaIO_4_ to produce BNPs. It has been demonstrated that the aldehyde groups on the hyperbranched polyglycerol (HPG) surface account for the adhesiveness of BNPs [17,18,19,20]. After Tris solution incubation, the aldehydes on the surface of the BNPs reacted immediately with the amine groups in the Tris structure to produce a Schiff base. The Schiff base protected the biological molecules from adhesion. At the same time, incorporation of the small-molecule drug IDB into the hydrophobic core of PLA-HPG was observed during the assembly process, yielding IDB/Tris-BNPs.

Dynamic light scattering (DLS) demonstrated that the diameters of IDB/NNPs, IDB/BNPs, and IDB/Tris-BNPs were almost the same. The hydrodynamic sizes of IDB/NNPs, IDB/BNPs, and IDB/Tris-BNPs were 122 nm, 120 nm, and 142 nm, respectively (Figure 2A). Due to the Tris coating on the surface of the IDB/Tris-BNPs, their average diameters were marginally larger than those of IDB/NNPs and IDB/BNPs. The polydispersity indices (PDIs) of IDB/NNPs, IDB/BNPs, and IDB/Tris-BNPs were 0.29, 0.26, and 0.19, respectively. The electrokinetic potential of the three nanoparticles (NPs) was above −20 mV (Figure 2B). In addition, the morphology of the NPs was observed by transmission electron microscopy (TEM), which indicated no considerable structural changes during NaIO_4_ oxidation and Tris incubation (Figure 2C). The three NPs were comparable in size and shape to the nifedipine-loaded BNPs (NFDP) obtained in a previous study [26].

Additionally, the encapsulation efficiency of the IDB/NNPs was approximately 51.52%, while those of IDB/BNPs and IDB/Tris-BNPs were approximately 40.95%. The relatively low encapsulation efficiency of IDB/BNPs may be attributed to mild drug leakages during the oxidation process. Furthermore, the drug release profiles of IDB/NNPs, IDB/BNPs, and IDB/Tris-BNPs were similar; all formulations slowly released the active ingredient until 96 h, with a total drug release rate of 70–80% (Figure 2D).

### 3.2. In Vitro Cell Viability Evaluation

The biocompatibility of IDB/NNPs, IDB/BNPs, and IDB/Tris-BNPs with skin cells was investigated. To determine the optimal application dose, mouse embryonic fibroblast cells (NIH/3T3 cells), B16 melanoma F10 cells (B16F10 cells), and human keratinocyte epithelial cells (HaCaT cells) were incubated with IDB/NNPs, IDB/BNPs, IDB/Tris-BNPs, free IDB, blank NNPs, blank BNPs, and blank Tris-BNPs at different concentrations of IDB for 24 h in vitro.

As shown in Figure 3A,C,E, there were no significant differences in the cytotoxicity of IDB between the different cell types. However, the viability of the NIH/3T3, B16F10, and HaCaT cell lines decreased with increasing IDB concentration. Based on these findings, 4 μg/mL was used as the optimal concentration of IDB in the subsequent experiments. We found that the cell viability of NIH/3T3, B16F10, and HaCaT cells was maintained after incubation with the NPs at concentrations within the 5–100 μg/mL range (Figure 3B,D,F). It was concluded that blank NPs were non-cytotoxic and biocompatible as nanocarriers. The cytotoxicity of NPs loaded with IDB may be attributed to the cytotoxicity of the released IDB. In addition, three blank PLA-HPG-based NPs exhibited considerable biocompatibility relative to polyethylene glycol (PEG)-based NPs, which have been the gold standard for several decades [29].

### 3.3. IDB/Tris-BNPs Reduced Melanin Synthesis after ɑ-MSH Stimulation of B16F10 Cells

To assess the effect of various NPs on the production of melanin, B16F10 cells, blank Tris-BNPs, IDB/NNPs, IDB/BNPs, IDB/Tris-BNPs, and commercial products were added to B16F10 cells before incubating with ɑ-MSH for 48 h. The melanin levels in the B16F10 cells decreased after treatment with IDB/NNPs, IDB/BNPs, IDB/Tris-BNPs, and commercial products, while no significant differences in melanin levels were observed between groups (Figure 4). In addition, the melanin levels of the blank Tris-BNPs and control groups were comparable. Therefore, the reduction in melanin synthesis induced by IDB/Tris-BNPs, IDB/NNPs, and IDB/BNPs was mainly due to the loading of IDB in the NPs.

### 3.4. Microneedle-Assisted Skin Penetration of NPs

Vehicles designed for optimal dermal drug delivery must facilitate the penetration of active pharmaceutical ingredients into the active layers of the skin. All NPs were loaded with PLA-Cy5 for fluorescence detection, to investigate the penetration of our Tris-BNP delivery strategy. The ability of NNPs, BNPs, and Tris-BNPs to penetrate the skin with and without the use of microneedles was compared. On day 4, the fluorescence signals emitted by NNPs, BNPs, and Tris-BNPs nearly disappeared in the absence of microneedle administration (Figure 5A,B). However, four days after microneedle-mediated topical administration, no NNPs and BNPs were detected; we found that Tris-BNPs diffused into the basal layer and remained within the skin (Figure 5C,D). These results revealed that, unlike BNPs, Tris-BNPs penetrated the skin after microneedle-mediated topical administration and were retained inside the epidermal layer, suggesting that Tris-BNPs could prolong IDB release due to prolonged NPs retention (Figure 5). With the help of microneedles, Tris-BNPs can penetrate the basal layer of normal skin, not only psoriatic skin whose SC has been broken [22].

### 3.5. Retention of Various NPs within Mice Skin following Pretreatment with Microneedles

The retention of PLA-Cy5-loaded NNPs, BNPs, or Tris-BNPs in skin tissues after microneedle-mediated topical administration was evaluated. We found that the fluorescence intensity due to Tris-BNP treatment was higher than that due to NNP and BNP treatments at 24 h, 48 h, and 72 h. Unlike NNPs and BNPs, Tris-BNPs exhibited fluorescence at 96 h (Figure 6A,B). Microneedle-mediated topical administration of Tris-BNPs significantly prolonged the retention of NPs in the skin, primarily attributed to the benefits of Tris-BNPs, harnessing the advantages of both NNPs and BNPs. The reduced fluorescence intensity of NNPs could be attributed to skin cell turnover or perspiration, as NNPs do not adhere to the epidermis. The non-bioadhesive nature of NNPs allowed them to penetrate the basal layer of the skin after microneedle-mediated administration, instead of remaining attached to the skin surface. BNPs, on the other hand, were restricted to the application site on the skin surface and could not penetrate it. In summary, these findings demonstrated that Tris-BNPs easily penetrated the SC following microneedle-mediated topical administration and interacted with amines in the underlying tissue layer, prolonging the retention of Tris-BNPs in the skin of C57BL/6 mice. The 96 h retention time indicates the advantages of applying Tris-BNPs with the aid of microneedles relative to recent studies, in which the majority of retention times were approximately 24 h [30,31,32].

### 3.6. Skin Protective Effect of NPs Administered by Microneedles to C57BL/6 Mice Skin after UV Irradiation

We evaluated the efficacy of various NPs delivered through microneedles in protecting the skin of C57BL/6 mice from UV radiation using parameters such as chroma values, melanin levels, ROS levels, and advanced glycation end products (AGEs) levels.

A skin colorimeter (IMATE M6601) was used to measure the chroma values, which indicate skin color. The Δchroma values were computed by deducting the chroma values measured on day 4 from the corresponding values obtained on day 1 after treatment with various NPs. Larger chroma values indicated a darker skin color, whereas smaller values indicated a lighter skin color. By comparing the chroma values of the UV and non-UV control groups, we found that the skin color darkened after UV radiation exposure, indicating that the artificial tanning model was successfully established. IDB/Tris-BNP treatment brightened the skin color, whereas IDB/NNPs, IDB/BNPs, and commercial product treatments alleviated skin discoloration (Figure 7A). Notably, mice in the IDB/Tris-BNPs group had lower melanin levels than mice in the UV and non-UV control groups, as well as the IDB/NNPs, IDB/BNPs, and commercial product-treated groups (Figure 7B). Therefore, IDB/Tris-BNPs exhibited a potent anti-melanogenic effect.

In vivo measurements of UV-induced ROS production demonstrated the antioxidant capacity of IDB/Tris-BNPs, IDB/NNPs, IDB/BNPs, and the commercial products. Mice in the UV control group had significantly increased ROS levels relative to mice in the non-UV control group, as well as the IDB/NNP, IDB/BNP, and commercial product-treated groups (Figure 7C). Treatment with IDB/Tris-BNPs reduced ROS levels to levels comparable to the non-UV control group. Notably, IDB/Tris-BNPs exhibited a stronger anti-ROS effect than commercial products. The effect of IDB/Tris-BNPs on the glycation process was evaluated by enzyme-linked immunosorbent assay (ELISA) after UV exposure. Treatment with IDB/Tris-BNPs significantly decreased AGEs levels compared to the IDB/NNP, IDB/BNP, and commercial product treatment after UV irradiation (Figure 7D).

### 3.7. Assessments of the Toxicity of NPs

We examined the toxicity of blank Tris-BNPs, IDB/NNPs, IDB/BNPs, IDB/Tris-BNPs, and commercial products using hematoxylin and eosin (H&E)-stained slices from various organ tissues (heart, liver, spleen, lung, kidney, and skin). No significant differences were observed in the tissues after treatment with blank Tris-BNPs, IDB/NNPs, IDB/BNPs, IDB/Tris-BNPs, and commercial products, indicating the biosafety of these NPs (Figure 8). Additionally, the intensity of the UV irradiation was moderate, as it only induced minor alterations and did not result in any structural modifications, as evidenced by a comparison of the H&E-stained sections from the UV control group with those from the non-UV control group.

Overall, our results demonstrated that administering IDB/Tris-BNPs with microneedles is a safe and effective strategy for preventing UV-induced skin damage, with the added benefits of desired skin diffusion and prolonged bioadhesion, as well as sustained IDB release.

## 4. Conclusions

Our study demonstrated that IDB/Tris-BNPs effectively reached the basal layer of the skin and were retained for at least 4 days, unlike NNPs and BNPs of similar sizes, which experienced a significant reduction in concentration within 24 h. Furthermore, BNPs were retained in the skin without effective penetration into the basal layer following microneedle administration. Notably, IDB/Tris-BNPs improved skin pigmentation, skin oxidation, and photoaging after a single topical application. IDB/Tris-BNPs showed superior performance against UV-induced damage relative to IDB/NNPs and IDB/BNPs. This effect was demonstrated by lower levels of ROS and AGEs in skin tissues, as well as suppressed melanogenesis. Therefore, the microneedle-assisted topical delivery of Tris-BNPs provides a promising platform for skin protection and disease treatment.

The excellent ability of IDB to protect the skin from UV damage has made it a first-line ingredient in skin care products. However, the dermal retention of IDB in conventional formulations can be affected by common disturbances such as sweat or swimming, making daily use necessary to maintain effective levels. Herein, we propose a microneedle-assisted sustained IDB delivery system that allows for direct access to the basal layer of the skin, which is most susceptible to damage. Unlike conventional formulations, which must penetrate almost the entire epidermis to reach the basal layer, our system achieves a sustained, long-term IDB release by employing IDB-loaded BNPs. The bioadhesivity of the BNPs was temporarily masked with Tris coating, which allowed them to diffuse through the epidermis before reaching the basal layer, where the bioadhesivity was recovered. IDB was gradually released from the hydrophobic core of the NPs to provide its skin-protecting effects. Our microneedle-assisted sustained IDB delivery system represents a promising strategy to improve the efficacy and prolong the duration of action of IDB.

## Figures and Tables

**Figure 1 biomedicines-11-01649-f001:**
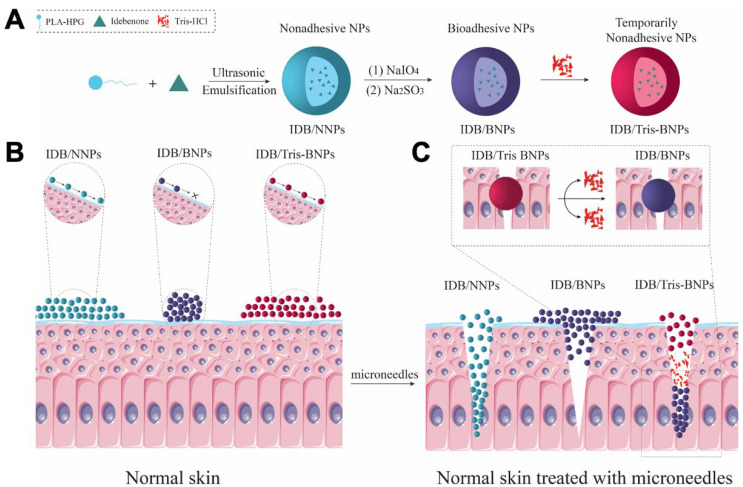
Preparation and topical application of (tris(hydroxymethyl)-aminomethane)-modified bioadhesive nanoparticles (Tris-BNPs). (**A**) The synthesis of idebenone (IDB)-loaded Tris-BNPs (IDB/Tris-BNPs) from IDB-loaded non-bioadhesive nanoparticles (IDB/NNPs) and IDB-loaded bioadhesive nanoparticles (IDB/BNPs). (**B**) IDB/NNPs, IDB/BNPs, and IDB/Tris-BNPs were applied onto normal skin without microneedles. (**C**) IDB/NNPs, IDB/BNPs, and IDB/Tris-BNPs were applied onto normal skin via microneedles.

**Figure 2 biomedicines-11-01649-f002:**
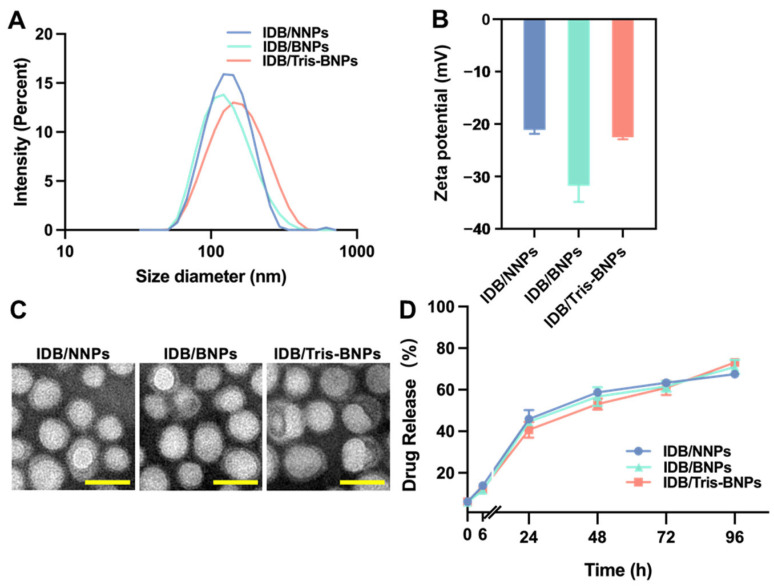
In vitro evaluation of nanoparticles. (**A**) Hydrodynamic diameter size distribution of different nanoparticle formulations. (**B**) Zeta potential of IDB/NNPs, IDB/BNPs, and IDB/Tris-BNPs. (**C**) Transmission electron microscopy (TEM) images of IDB/NNPs, IDB/BNPs, and IDB/Tris-BNPs. (Scale bars = 200 nm) (**D**) Drug release rate (%) of IDB/NNPs, IDB/BNPs, and IDB/Tris-BNPs at 37 °C. All experiments were carried out in triplicate. Data are shown as mean ± s.d.

**Figure 3 biomedicines-11-01649-f003:**
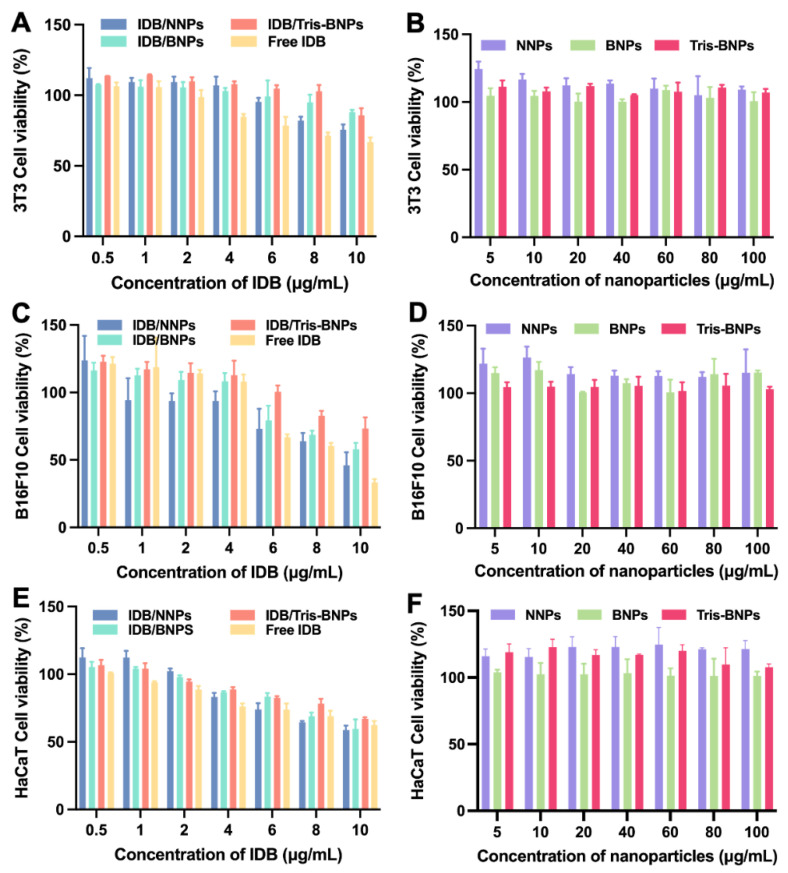
Cell viability of NIH/3T3, B16F10, and HaCaT cells incubated with IDB/NNPs, IDB/BNPs, IDB/Tris BNPs, free IDB, blank NNPs, blank BNPs, and blank Tris BNPs for 24 h. (**A**,**C**,**E**) The cell viability of NIH/3T3, B16F10, and HaCaT cells treated with IDB (0.5 to 10 µg/mL)-loaded NPs was assessed using a Cell Counting Kit (CCK)-8 assay. (**B**,**D**,**F**) The cell viability of NIH/3T3, B16F10, and HaCaT cells treated with blank NNPs, BNPs, and Tris BNPs (5 to 100 µg/mL) was measured using the CCK-8 assay. All experiments were carried out in triplicate. Data are expressed as mean ± s.d.

**Figure 4 biomedicines-11-01649-f004:**
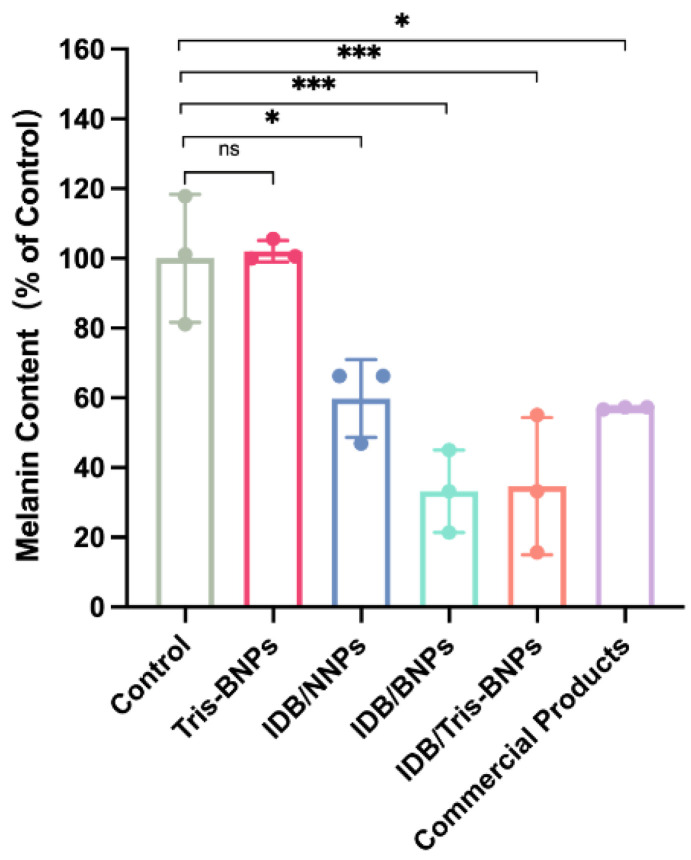
Effect of various NPs on melanogenesis in alpha-melanocyte-stimulating hormone (ɑ-MSH)-stimulated B16F10 cells. All experiments were carried out in triplicate. Data are expressed as mean ± s.d. * *p* < 0.05, *** *p* < 0.001.

**Figure 5 biomedicines-11-01649-f005:**
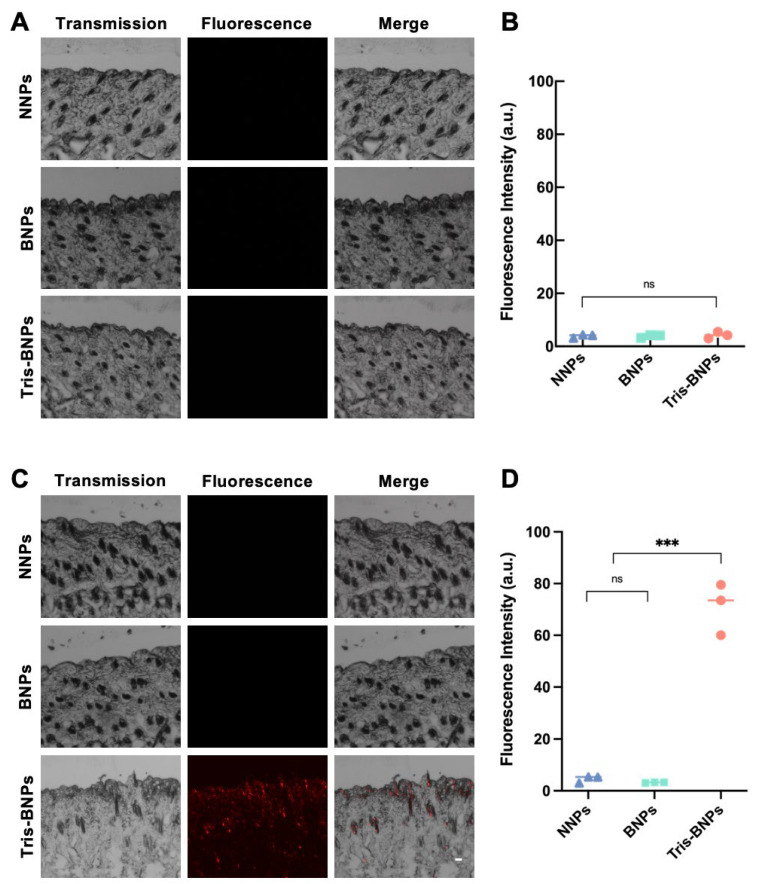
Evaluation of Tris-BNPs adhesion and penetration with or without microneedle-mediated topical administration in healthy C57BL/6 mice. (**A**) NNPs, BNPs, and Tris-BNPs loaded with PLA-Cy5 at 1.0 mg/mL were topically applied to healthy C57BL/6 mice, without the use of microneedles. (**B**) The fluorescence intensity of mice skin without microneedle-mediated topical administration was quantified in each group. All experiments were carried out in triplicate. Data are expressed as mean ± s.d. (**C**) NNPs, BNPs, and Tris-BNPs at 1.0 mg/mL were administered to the healthy skin of C57BL/6 mice using microneedles. (Scale bars = 50 μm) (**D**) The fluorescence intensity of mice skin was quantified in each group. All experiments were carried out in triplicate. Data are expressed as mean ± s.d. *** *p* < 0.001.

**Figure 6 biomedicines-11-01649-f006:**
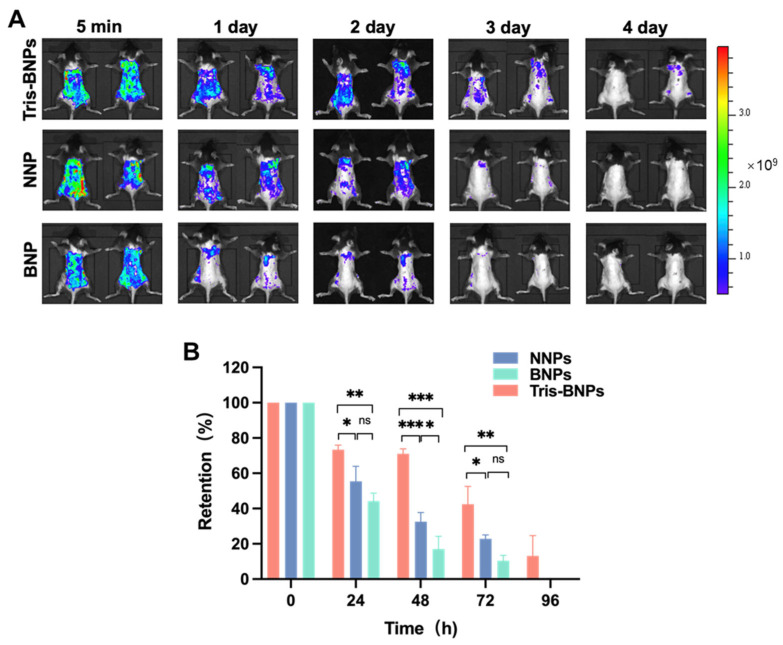
Evaluation of NP retention. (**A**) NNPs, BNPs, and Tris-BNPs encapsulating PLA-Cy5 were applied on the normal skin of C57BL/6 mice with microneedles. The fluorescence signal retention was imaged with Xenogen at different time points. (**B**) The fluorescence intensity was quantified and normalized to the baseline intensity in each group. All experiments were carried out in triplicate. Data are expressed as mean ± s.d. * *p* < 0.05, ** *p* < 0.01, *** *p* < 0.001.

**Figure 7 biomedicines-11-01649-f007:**
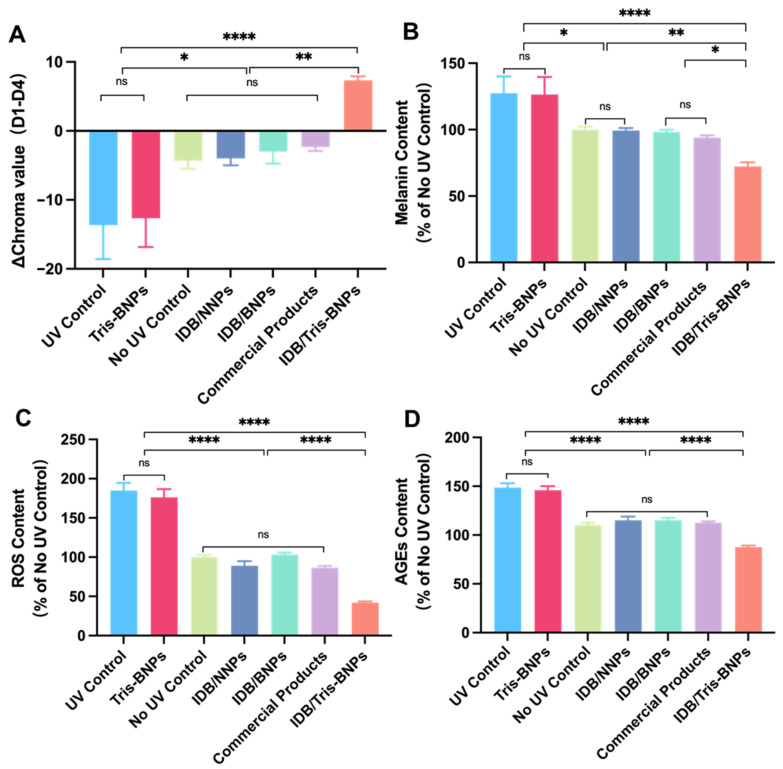
Skin protection effect of NPs administered with microneedles to C57BL/6 mice skin after UV irradiation. (**A**) The changes in chroma values between day 1 and day 4 in each group. (**B**) The melanin content (%) on day 4 in each group. Statistical differences were compared with the non-UV control group. (**C**) The ROS content (%) on day 4 in each group. Statistical differences were compared with the non-UV control group. (**D**) The AGEs content (%) on day 4 in each group. Statistical differences were compared with the non-UV control group. Data are expressed as mean ± s.d. (n = 3). * *p* < 0.05, ** *p* < 0.01, and **** *p* < 0.0001.

**Figure 8 biomedicines-11-01649-f008:**
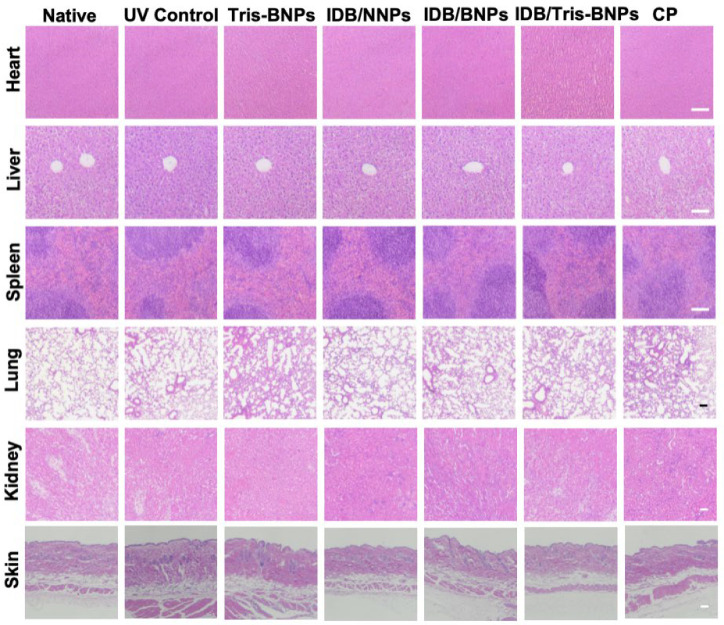
In vivo toxicity evaluation. Hematoxylin and eosin (H&E) staining images of major organs of C57BL/6 mice on day 4 after various NP treatments. CP, commercial product (Scale bar = 100 µm).

## Data Availability

The data are available from the corresponding author.

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
