# Peer review of "Microneedle-Assisted Topical Delivery of Idebenone-Loaded Bioadhesive Nanoparticles Protect against UV-Induced Skin Damage"

_biomedicines, 2023, doi:10.3390/biomedicines11061649_

Round 1

Reviewer 1 Report

1- The abstract must rewrite as there is no numerical result added to this section. Try to add some important results of the study to the abstract.

2- The title is not clear. It is suggested to rewrite the topic simply for better understanding.

3- To improve the introduction, the following references give enough information about the microneedles and their functions:

Sabbagh, F., & Kim, B. S. (2023). Ex Vivo Transdermal Delivery of Nicotinamide Mononucleotide Using Polyvinyl Alcohol Microneedles. Polymers15(9), 2031.

Sabbagh, F., & Kim, B. S. (2022). Microneedles for transdermal drug delivery using clay-based composites. Expert Opinion on Drug Delivery19(9), 1099-1113.

4- In the last paragraph of the introduction, the novelty and aim of the study must be added. 

5- The current last paragraph is discussing the results, shift this paragraph from the introduction to the conclusion.

6- If the following sentence is the authors finding, why there is a reference at the end? "Therefore, we hypothesized that Tris-BNPs could be applied to normal skin lacking the high permeability characteristics of psoriasis, given that melanocyte in the epidermis are typically located in the basal layer above the basement membrane [23]."

7- All the results must be compared with other similar studies.

Extensive editing of the English language is required.

Reviewer 2 Report

Dear authors

Let me congratulate you for  such an interesting article. The idea to incorporate needles that are sensitive to media is very original. The techniques developed are coherent with your discussions. I have enjoyed with the mechanisms shown to obtain the NP's. The size distribution, very accurate and the absorption studies, very interesting.

In my opinion, the article is ready to be published

Many thanks for your work

Author Response

We thank for the reviewer's kind recommendation and encouragement. Hope everything goes well with you.

Best wishes.

Reviewer 3 Report

The present manuscript revealed that IDB delivery strategy could improve the long-term protective effects against UV-induced skin damage. However ,there are several mistakes should be clarified. 

1. In vitro drug release test must be carried out to prove whether nano IDB can effectively release IDB to achieve the effect.

2. The preparation of Tris-BNPs, IDB/NNPs, IDB/BNPs and/or IDB/Tris-BNPs used organic solvents, such as DMSO, ethyl acetate, diethyl ether and  NaIO4, to synthesize the IDB nanoparticle and microneedle. The residue organic solvent must be detected.  

3.  Section 3.6. Biocompatibility evaluation of NPs is an incorrect name for the experiment. This test is for in vitro cytotoxicity. All samples shall be prepared according to the guidelines of ISO-10993 to prepare the test samples. Because the microneedles designed in this paper belong to medical device. In vitro cytotoxicity assays require the addition of pictures of cell morphology to illustrate the cellular safety of the sample.

4. The results in the manuscript all illustrate how NanoIDB or microneedle reduces changes in melanin and chromatic aberration after UV exposure. This is inconsistent with the NanoIDB theme of preventing UV skin damage.

5. Why did authors not analyze the IDB content in mouse skin?

6. The epidermis thickness in UV treatment was obviously thicker than native group. Why did authors not determine the epidermis thickness for elucidating the UV-induced Skin Damage? In addition, how about the sunburn cell amount?

Minor mistakes:

Page 11,  The numbers for Na2SO3 and NaIO4 are required subscripts.

Reviewer 4 Report

This study investigated Tris (tris(hydroxymethyl)-aminomethane) modified bioadhesive nanoparticles (Tris-BNPs) loaded with Idebenone (IDB) were prepared to test the penetration into the skin and the protection of UV-induced skin damage. The results indicated that it could diffuse from the skin surface to the basal layer and Tris-BNPs diffused during the penetration process, resulting in the exposure of aldehyde groups in bioadhesive nanoparticles that could react with amine groups in the basal layer cell membrane protein. The nanoparticles stayed for more than 4 days and released IDB. The nanoparticles treated with microneedle exhibited anti-melanin and reduced UV-induced skin damage.

1.          The abstract should include all the results of the work; however, it did not describe all the results of this study. Authors have to reorganize and rewrite their abstract.

2.          Idebenone exhibited the effect on antioxidant and skin-aging protection activities. However, this study investigated the effects of IDB/Tris-BNPs, IDB/NNPs and IDB/BNPs, but not idebenone. The experiments is incomplete and must be redoned.

3.          Authors stated that “n this respect, Tris (tris(hydroxymethyl)-aminomethane) modified bioadhesive nanoparticles (Tris-BNPs) loaded with IDB were prepared that could diffuse from the skin surface to the basal layer. Instead of instant adhesion, once Tris-BNPs were applied to the epidermis, the Tris coating gradually diffused during the penetration process, resulting in the exposure of aldehyde groups in bioadhesive nanoparticles that could react with amine groups in the basal layer cell membrane protein.” . However, they did not prove this resulted. More experiments were needed in this work.

4.          Figure 2c, the scale bar have to label clearly in the Figure.

Minor editing of English language was required.

Round 2

Reviewer 1 Report

The comments are addressed correctly and the paper is accepted in the current form.

Reviewer 4 Report

Authors have revised their manuscript according to the comments.

The format of references should follow the format of Biomedicines.